# Clinical Utility of Genomic Tests Evaluating Homologous Recombination Repair Deficiency (HRD) for Treatment Decisions in Early and Metastatic Breast Cancer

**DOI:** 10.3390/cancers15041299

**Published:** 2023-02-18

**Authors:** Loïck Galland, Nicolas Roussot, Isabelle Desmoulins, Didier Mayeur, Courèche Kaderbhai, Silvia Ilie, Audrey Hennequin, Manon Reda, Juliette Albuisson, Laurent Arnould, Romain Boidot, Caroline Truntzer, François Ghiringhelli, Sylvain Ladoire

**Affiliations:** 1Department of Medical Oncology, Georges-François Leclerc Center, 21000 Dijon, France; 2Platform of Transfer in Biological Oncology, Georges-François Leclerc Cancer Center, 21000 Dijon, France; 3University of Burgundy-Franche Comté, 21000 Dijon, France; 4Department of Pathology and Tumor Biology, Georges-François Leclerc Center, 21000 Dijon, France; 5Research Center INSERM LNC-UMR1231, 21000 Dijon, France; 6ICMUB UMR CNRS 6302, 21000 Dijon, France; 7Bioinformatic Core Facility Georges-François Leclerc Cancer Center, 21000 Dijon, France; 8Genomic and Immunotherapy Medical Institute, Dijon University Hospital, 21000 Dijon, France

**Keywords:** breast cancer, early breast cancer, metastatic breast cancer, *BRCA*, NGS, HRD score, homologous recombination deficiency, mutational signature, PARPi, platinum salts

## Abstract

**Simple Summary:**

Breast cancer is the most frequently occurring cancer worldwide. With the help of next-generation sequencing, the development of biomedical technologies and the use of bioinformatics, it is now possible to identify specific molecular alterations in tumor cells, such as homologous recombination deficiencies, enabling us to consider using DNA-damaging agents such as platinum salts or PARP inhibitors. In this review, we summarize current knowledge on the clinical utility of genomic tests evaluating homologous recombination repair deficiency for treatment decisions in early and metastatic breast cancer.

**Abstract:**

Breast cancer is the most frequently occurring cancer worldwide. With its increasing incidence, it is a major public health problem, with many therapeutic challenges such as precision medicine for personalized treatment. Thanks to next-generation sequencing (NGS), progress in biomedical technologies, and the use of bioinformatics, it is now possible to identify specific molecular alterations in tumor cells—such as homologous recombination deficiencies (HRD)—enabling us to consider using DNA-damaging agents such as platinum salts or PARP inhibitors. Different approaches currently exist to analyze impairment of the homologous recombination pathway, e.g., the search for specific mutations in homologous recombination repair (HRR) genes, such as *BRCA1*/*2*; the use of genomic scars or mutational signatures; or the development of functional tests. Nevertheless, the role and value of these different tests in breast cancer treatment decisions remains to be clarified. In this review, we summarize current knowledge on the clinical utility of genomic tests, evaluating HRR deficiency for treatment decisions in early and metastatic breast cancer.

## 1. Background

Breast cancer is the most frequently occurring cancer in the world, with increasing incidence, and it is becoming a major public health problem [1]. It is therefore increasingly important to identify tools that can guide physicians in their therapeutic choices, both at the localized and metastatic stages. Among these tools, the evaluation of the homologous recombination (HR) process could prove to be of interest. Its clinical utility and its current place in the breast cancer landscape are the subject of this review.

### 1.1. Repair of DNA Double-Strand Breaks and Homologous Recombination (HR) Deficiency

DNA double-strand breaks (DSBs) may be linked to physiological (e.g., during meiosis or telomere erosions) and/or pathological mechanisms [2,3]. These pathological mechanisms may be the consequence of replication accidents or may result from the action of exogenous agents (such as radiotherapy or chemotherapy). If DSBs accumulate, the cell becomes non-viable and dies. Various pathways are involved in DNA repair, when DSBs arise [4].

First, non-homologous end joining (NHEJ) or micro-homologous end joining (AltEJ) pathways are active throughout the cell cycle and enable rapid but error-prone repair [5]. The second important pathway, called “homologous recombination repair (HRR) pathway”, is the only one able to repair double-stranded DNA lesions *ad integrum*. This pathway involves several key proteins such as BRCA1 and BRCA2 but also many other actors such as RAD50, RAD51, or PALB2. During this process of HR, DSBs are recognized by the MRN complex (Mre11-Rad50-Nbs1), which transforms the double-stranded ends into single strands [6]. These single strands are initially passively coated with an RPA protein, and BRCA2 will replace RPA with the RAD51 protein [7]. The main steps and proteins involved in this HR process are summarized in Figure 1.

This pathway can be inactivated by numerous somatic events (mutations, deletions, and methylation of the promoters of the genes involved), with or without associated germline mutations in many solid tumors (breast, ovary, pancreas, prostate, and stomach or lung tumors) [8]. Deficiency of the HR pathway represents a mechanism of oncogenesis, increasing genetic instability, and promoting the activation of oncogenes and the inactivation of tumor suppressor genes. This is known as homologous recombination deficiency (HRD). As explained above, *BRCA1* and *2* genes are considered to be tumor suppressor genes, and their inactivation is responsible for a predisposition to breast or ovarian cancer [9]. This HR deficit is frequently found in high grade ovarian cancers and in breast cancers (BC). It is estimated that 70–80% of breast cancer patients with a *BRCA1* or *2* mutation have a TNBC subtype, and that about 20% of TNBC have a *BRCA1* or *2* mutation [10]. Approximately 10–36% of BCs that occur in *BRCA1*/*2* mutation carriers are estrogen-receptor (ER)-positive (ER+) [11]. Sometimes, the mutations found in this pathway do not affect *BRCA 1/2* but rather other genes involved, also leading to genomic instability and to a “*BRCA*ness or HRness phenotype” (such as *RAD51*C epimutations, inactivation of *PALB2*, *BRIP1*, or *BARD1*) [12]. Mutations caused by malfunction of the HR process occur in a specific pattern, or “signature”. This mutation profile in cancer DNA thus appears to be a good way to identify breast cancers with a defect in HR DNA repair, regardless of the underlying cause [13].

### 1.2. Homologous Recombination Deficiency: Therapeutic Interest

The deficiency of the homologous recombination pathway also represents an “Achilles heel” of the tumor, with the development of molecules that take advantage of this inactivation (PARP inhibitors and chemotherapy with platinum salts in particular) [14,15,16]. Indeed, these molecules are able to create numerous DSBs in the DNA, which can no longer be repaired in cancer cells that are highly deficient in HR. The assessment of HRD status and the therapeutic value of treatments affecting this pathway initially originated in ovarian cancer.

Platinum salts are cytotoxic chemotherapies that induce binding of alkyl groups on the purine bases of DNA, enabling the creation of mono- or bi-functional adducts and intra- and/or inter-strand bridges [17,18]. The result is to halt the cell’s transcription and replication processes. The HR pathway is required to repair platinum-induced double-strand breaks, which explains the greater sensitivity of HRD tumors to this therapeutic class. Thus, sensitivity to platinum salts could be considered indirectly as a possible clinical marker of tumor HRD.

In cells with inactivating mutations of the *BRCA1*/*2* genes, the HR pathway is deficient, and survival of these tumors relies on one or more accessory repair pathways. Poly-(ADP-ribose) polymerases (PARP) are enzymes that induce synthesis of a poly-ADP ribose chain, acting as a signal to initiate repair in the base excision repair (BER) pathway. PARP inhibitors (PARPi) are compounds that trap PARP on sites of DNA damage, leading to replication fork stalling and to the generation of DSBs, resulting from unresolved SSBs [19]. Thus, this accumulation of DSBs that cannot be repaired in HR-deficient cells leads to cell cycle arrest in G2/M and to apoptosis of the tumor cells. This phenomenon is now well known as “synthetic lethality” [14,16,20,21]. Olaparib, a selective PARP-1 inhibitor, was initially developed in advanced, high-grade, relapsing, platinum-sensitive ovarian cancer [22]. In addition, while the first trials and registrations only concerned *BRCA1* or *2* mutations, other trials have explored the extension of indications to tumors that are *BRCA1*/2 wild-type (WT), but that are considered to harbor HRD.

### 1.3. Tools to Assess Homologous Recombination Deficiency in Tumors

Given the therapeutic challenges of identifying platinum salt or PARPi-sensitive tumors, a number of biological tools have been developed, primarily to detect *BRCA*-mutated tumors, but also to identify HRD tumors outside the context of *BRCA* mutations [23,24].

The first approach being developed to this end is the search for mutations in HR pathway genes [25]. Beyond germline *BRCA* mutations, there seems to be evidence of the value of identifying somatic exclusive mutations, notably in prostate [26] and ovarian [27] cancer. In breast cancer, the results of the TBCRC 048 study seem to confirm the potential interest of identifying exclusive *BRCA* somatic mutations to predict response to PARPi [28]. The clinical and therapeutic relevance of the detection of mutations other than *BRCA 1* or *2* seems to depend on the histological type of the cancer. For example, in the previously mentioned TBCRC 048 study, germline mutations in *PALB2* were also associated with response to PARP inhibitors. The results seemed interesting for some, but not all of these genes, raising the question of the panel of genes other than *BRCA* to study in each cancer.

An alternative method consists in the use of a genomic profile (or genomic signature) that reflects HRD in tumor cells, regardless of its molecular origin [24,29]. Indeed, DNA-based measures of genomic instability capturing large genomic aberrations (“genomic scars”) resulting from HRD have been developed in recent years, and represent an alternative approach for identifying HR-deficient tumors. Three independent scores have been developed: The Curie Institute developed a profile based on the number of chromosomal status changes (or breaks), and more specifically, on breaks in large chromosomal regions >10 Mb (Large-scale state transition, LST). This profile was initially identified in TNBC, since Popova et al. showed that the number of LSTs was significantly associated with *BRCA1* inactivations in this tumor subtype [30]. Another team showed that an allelic imbalance in subtelomeric regions (Telomeric Allelic Imbalance, TAI) was significantly associated with platinum sensitivity in TNBC as well as in *BRCA* WT ovarian tumors [31]. *BRCA1* or *2* mutated tumors were also more likely to develop loss of heterozygosity (LOH). An overall measurement of allelic balance, with detection of large regions of LOH, which seems to correlate well with the deficit of HR has been developed under the name of “FoundationFocus CDx” [32]. This HRD-LOH profile is based on the detection of large regions (>15 Mb) of heterozygosity loss, and enables the detection of *BRCA* mutations in ovarian cancer.

Timms and colleagues subsequently demonstrated that the combined presence of LST, TAI, and LOH across the genome seems to be of even greater value in predicting HRD status, leading to the commercialization of a combined score called “myChoice HRD” (Myriad genetics) [33]. The assay yields a “HRD score”, considered to be positive if the score is ≥42 (cutoff value validated in ovarian cancer). This score is currently the most widely used in the world. Moreover, as in ovarian cancer, this combined score could help to predict sensitivity to molecules that take advantage of the HR defect, such as platinum salts and PARPi [34]. However, it is important to note that these genomic profiles measure are established early in tumorigenesis. This profile will therefore persist during tumor progression, leading to the term “genomic scars”. Thus, the profile provides valuable information about the initial HR status of the tumor, but not necessarily about the current status, especially at advanced stages of disease. HRD status (from a functional point a view) can evolve over time, with partial or complete restoration of HR pathway functionality, most often under the therapeutic pressure of HRD-targeting agents, and secondary mutations restoring HR function appear to be a mechanism of resistance [35,36].

Other tools are also available to assess the HR status of a tumor. Following the analysis of different mutations found in thousands of exomes from different tumors in the TCGA (The Cancer Genome Atlas) or the ICGC (International Cancer Genome Consortium), “mutational signatures” have been defined and referenced in the Catalogue of Somatic Mutations In Cancer (COSMIC) [37]. These different mutational profiles are characteristic patterns created on the cancer cell genome by each mutational process. Next-generation sequencing was used to obtain the mutational spectrum of these tumors, leading to the categorization into specific signatures [38]. In particular, signature 3 has been found to be predominantly expressed in breast or ovarian cancers and linked to a defect in HR [38,39,40] and response to platinum salts. However, all *BRCA* 1 or *BRCA* 2 pathogenic mutations do not result in a single mutational signature: other signatures [41] also seem to be associated with a deficit of HR, such as signature 8 for example [38,42,43]. Based on emerging knowledge of these mutational signatures, different algorithms have been developed to help define the HRD status of a given tumor, such as SignatureAnalyzer [44] or Signature Multivariate Analysis (SigMA) [45].

Recent advances in sequencing technologies, with reduced overall costs, have prompted the development of tools based on whole genome sequencing (WGS), such as the HRDetect tool, which has been developed and presented as a predictive score of HRD. This score combines different mutational signatures (incorporating COSMIC signatures 3, 5, and 8), as well as other elements mentioned above such as microhomology-mediated deletions, TAI, LOH, and LST. With this score, Davies et al. were able to detect *BRCA* deficiency (germline and/or somatic) with a sensitivity of 99% in a cohort of 560 TNBC, and identified 47 tumors with a functional *BRCA* deficiency without any mutation found [42]. Accordingly, the number of tumors with HRD increased from 1–5% to 22%, leading to increasing numbers of patients who could potentially benefit from platinum salts or PARPi.

As HRD tumors may evolve towards restoration of HRR and acquire resistance to DNA-damaging agents, such tumors may be misclassified by genomic scar/signature-based assays. Thus so-called “functional” tests have also been developed. These tests dynamically evaluate the ability of tumor cells to perform the HR mechanism. For example, it is possible to measure the nuclear accumulation of RAD 51 [46] and tumors classified as RAD 51-low (by immunofluorescence [47] or immunohistochemistry [48]) would have a functional HRD. The interest of this functional test has been evaluated in various cancers, including breast [49,50,51] and ovarian cancer [47,52,53], and may be a biomarker for PARPi and/or platinum response. The most well-known test, the REcombination CAPacity (RECAP test) [47,49], classified tumors in three HR groups (deficient, intermediate, or proficient), depending on their RAD 51 score [54]. Despite some technical limitations [46,48,55], these tests have the advantage of providing an assessment of the current HR status of the tumor, to detect resistance acquired under therapeutic pressure, and to detect restoration of homologous recombination in initially HRD tumors [56]. The RAD 51 test has been retrospectively validated on cohorts from prospective clinical trials in ovarian [57], prostate [58], and triple-negative breast [59] cancers but require further clinical validation and standardization for routine use.

Clearly, it is important, especially in the field of breast cancer, to critically evaluate the validity and clinical utility of these HRD tests (DNA-based and/or functional). The main objective is to help to predict the sensitivity of tumors to DNA-damaging agents such as PARPi inhibitors or platinum salts, both in localized and metastatic situations. Figure 2 summarizes the different assay strategies discussed in this last section.

## 2. Early Breast Cancer (eBC)

Breast cancer represents nearly 30% of female cancers and patients harboring early stage disease are treated with a view to cure [1]. Nowadays, preoperative treatment is standard of care for a large proportion of early breast cancers enabling a down-staging and the assessment of treatment responsiveness, which is critical to adapting the adjuvant regimen. Evidence shows that pathologic complete response (pCR)—defined as the absence of infiltrative tumor cells in the breast and axilla (ypT0/is ypN0)—is associated with better outcomes, especially among the aforementioned aggressive subtypes [60]. Hence, achieving pCR with a preoperative regimen in this setting has become one of the main objectives of treatment in TNBC. Approximately 30–40% of TNBCs were shown to achieve pCR with a neoadjuvant cytotoxic regimen containing anthracyclines and taxanes [60,61]. Moreover, as previously mentioned, a majority of patients with *BRCA 1/2* mutations harbor this intrinsic subtype, and nearly a fifth of TNBC all-comers bear these mutations [10]. Given the relationship between TNBC and *BRCA 1/2* mutation, the use of additional therapies that target HRD—such as platinum salts or PARPi—is an attractive approach to improve the pathological response rate, and to achieve curative goals.

Here, we recap how evidence-based medicine has evolved in this setting, with the emergence of genomic tests evaluating HRD, to assist clinicians in treatment decision-making, and we review the clinical utility of these assays.

### 2.1. Platinum Salts and PARPi for BRCA-Mutated eBC

Firstly, the GEICAM/2006-03 trial [62] was the first randomized study to add platinum salts to standard neoadjuvant chemotherapy (NAC) in TNBC, regardless of *BRCA* status. Of the 94 patients included, 49 received carboplatin in addition to an anthracycline-taxane based regimen, but the results failed to demonstrate any benefit in terms of pathologic response rate. A few years later, the CALG-B 40603 [63] phase II trial enrolled early TNBC to assess the addition of carboplatin. In that study, Sikov et al. demonstrated that adding carboplatin to standard chemotherapy increased the pCR rate, which was achieved in more than half of patients (54% vs. 41%, *p* = 0.0029). However, additional carboplatin did not result in any benefit in terms of event-free or overall survival benefit [63]. Considering these data, there was a keen interest in refining the selection of patients who might benefit from the addition of platinum salt in the neoadjuvant setting. Because of the centrality of *BRCA* in the homologous recombination process, the role of platinum salts in *BRCA*-mutated patients was studied first.

Byrski et al. assessed a single NAC regimen with cisplatin in a small cohort patients only with *gBRCA1* alteration, of whom 76.6% had the triple-negative subtype, achieving a promising pCR rate of 61% [64]. Of note, although there was a small proportion of estrogen receptor-positive (ER+) disease, 56% of them achieved ypT0/is ypN0 after single-platinum chemotherapy. Based on these data, the same regimen was compared to the standard doxorubicin-cyclophosphamide chemotherapy in the randomized phase II INFORM trial (TBCRC 031) [65]. Here again, only *gBRCA* mutation carriers were enrolled. Unexpectedly, only a few patients achieved complete response, 18% in the cisplatin arm and 26% in the comparative arm (HR = 0.70, 90% CI [0.39; 1.2]). Results were similar in the TNBC and ER+ populations, although the number of ER+ patients was small. One hypothesis that may explain these findings is that patients included in the INFORM trial had more advanced stage disease than in the study by Byrski et al. Nevertheless, the results were subsequently corroborated in large, phase III trials evaluating different NAC regimens in TNBC, such as the GeparSixto [66] and BrighTNess [67] trials. These trials demonstrated higher pCR rates for patients receiving carboplatin in addition to a standard anthracycline/taxane NAC backbone for TNBC in the whole population. In BrighTNess [67], 634 patients with TNBC were enrolled, and the authors showed an increased pCR rate with the combination of platinum and PARPi (veliparib) compared to standard chemotherapy (53% vs. 31%, *p* < 0.0001), but not when compared to patients receiving only additional carboplatin, in whom a promising 58% pCR was achieved (*p* = 0.36). A post-hoc analysis confirmed that the benefit obtained with carboplatin alone was significant (*p* < 0.001). While reinforcing NAC with platinum offered a significantly improved pCR rate, it seems that veliparib addition did not yield any benefit. Thus, since these results, the combination of carboplatin and paclitaxel has become standard of care in NAC for TNBC all-comers. Interestingly, the odds of pCR were not higher in patients with *BRCA* mutations receiving carboplatin, or carboplatin + veliparib, when compared with matched non *BRCA*-mutated patients [68]. Later, the assumption that patients harboring *gBRCA* mutation do not benefit from platinum addition, contrary to *BRCA* WT patients, was confirmed in a meta-analysis encompassing more than 300 *BRCA*-mutated patients [69]. These intriguing findings could be explained by the excellent results obtained with standard chemotherapy (which already contains some DNA-damaging agents such as alkylants or anthracyclines) in *BRCA*-mutated cases [66,67,70].

It therefore seemed essential to evaluate other potentially valuable drugs in these *BRCA*-mutated patients with eBC. In particular, cumulating lines of evidence pointed to PARPi activity in advanced ovarian, prostate, pancreatic, and breast cancers harboring *BRCA1*/*2* mutation [71,72,73]. Moreover, in the original phase II I-SPY-2 trial, Rugo et al. estimated that a carboplatin-PARPi regimen on top of the standard anthracycline-taxane based chemotherapy, had an estimated probability of pCR of 51% in TNBC [74]. Nevertheless, with such a combination in the experimental arm, deciphering the effectiveness of each drug alone remains problematic. Therefore, two neoadjuvant trials aimed to assess the efficacy of a single PARPi regimen in the setting of *gBRCA 1/2* mutation, and confirmed substantial activity, with pCR rates reaching 49% and 40% with talazoparib [75] and niraparib [76] respectively. However, it should be mentioned that a significant proportion did not respond to PARPi monotherapy in these two studies, which means that this strategy cannot currently constitute a standard treatment compared to NAC.

Later, Tutt et al. designed the OlympiA trial, to assess the efficacy of PARPi therapy (olaparib for 1 year) in the adjuvant setting. This phase III study enrolled eBC with g*BRCA1*/*2* mutation carrying high-risk clinicopathological factors after definitive local treatment and neoadjuvant or adjuvant chemotherapy [77]. Results were in favor of olaparib in terms of invasive-free survival (HR = 0.58, 99.5% CI [0.41; 0.82]) which later translated into a significant overall survival benefit (HR = 0.68, *p* = 0.009) [78]. Although adjuvant capecitabine was not permitted in this trial (as in the CREATE-X trial [79]), therefore precluding direct comparison, data in the metastatic setting may suggest that olaparib is a better choice for g*BRCA* carriers harboring TNBC [80,81]. It is important to note that this study also included ER+ tumors, which may also benefit from this treatment. Accordingly, OlympiA is a practice-changing study that has demonstrated the clinical utility of g*BRCA* testing in this high-risk population of eBC.

### 2.2. Targeting BRCAness in eBC beyond BRCA1/2 Mutations

Whole genome sequencing analyses from a Swedish database revealed that among TNBC carriers harboring a high HRDetect mutational signature, 67% was explained by germline/somatic *BRCA1/2,* as well as by other genomic/epigenic abnormalities (*BRCA1* promoter hypermethylation, *RAD 51C* hypermethylation, or biallelic loss of *PALB 2*), illustrating the existence of many alternative alterations that may lead to HRD tumor status [42]. Patients with HRDetect-high tumors were also found to have a better invasive-disease-free survival after adjuvant chemotherapy than those with HRDetect low tumors.

A number of authors have assessed the HRD score in the setting of early TNBC [29,33]. Three neoadjuvant trials reported that genomic instability, reflected by an HRD-score ≥ 42 or *BRCA1*/*2* mutation significantly predicts pCR with NAC including platinum salts [29]. When restricted to the *BRCA* WT population, high-HRD score remains a predictor of response to platinum salts, demonstrating that an assay evaluating genomic instability may be able to identify a wider range of patients who might benefit from such a regimen, thus offering critical information for treatment decision-making. 

More recently, translational analyses from the phase II randomized TBCRC 030 study comparing neoadjuvant cisplatin to paclitaxel chemotherapy, examined the role of HRD biomarkers and their associations with response to NAC in this TNBC population [82]. The threshold of positivity of the HRD score to define tumors deficient for HR was found to be 33 (and not 42). The results did not support an association between the presence of HRD and better response to platinum. Results remained unchanged in exploratory analysis using the more common threshold of ≥42 as a cut-off for HRD positivity.

Moreover, further exploratory analyses conducted in the BrighTNess trial, assessing the prognostic and predictive value of HRD-score, showed that patients with HRD-high tumors (with a cutoff value of either ≥42 or ≥33) had higher pCR rates, whatever the neoadjuvant treatment received. Patients treated with additional carboplatin had higher pCR, both in the HRD-high and HRD-low subgroups, and the odds of pCR were not better in patients with HRD-high tumors receiving carboplatin, or carboplatin + veliparib, compared to patients with HRD-low tumor [83]. Similar results were observed in the TNBC population from the GeparSixto study; these authors found HRD-high scores in 70.5% of TNBC, of whom 60.3% had high-HRD score without *BRCA* mutation [84]. Here again, HR deficiency was an independent predictor of pCR, but did not predict carboplatin benefit. Taken together, these results suggest that HR deficiency evaluated by HRD score may be a predictor of response to NAC, but not of the benefit of carboplatin on top of standard NAC. Therefore, this evidence does not support routine clinical use of this genomic assay in such decision making.

Later, assessment of genomic instability focusing on RAD 51 foci was undertaken in the same GeparSixto trial [59]. RAD 51-low score, reflecting a functional HRD phenotype, was closely concordant with genomic HRD-score, with 87% accuracy. As a HRD genomic test, RAD 51 score is able to identify tumors without *BRCA* mutation harboring epigenetic or other HR gene alterations that are supposed to sensitize them to DNA-damage therapy. RAD 51-low tumors treated with carboplatin were more prone to achieve pCR, contrary to RAD 51-high tumors. Furthermore, contrary to the HRD-score, the RAD 51 assay independently predicts platinum benefit. These results support further development of this assay to guide decisions about whether to add a carboplatin to standard NAC or not.

Rather than combining platinum salts with PARPi (as in the BrighTNess trial), the GeparOLA study aimed to replace platinum with a PARPi in a HR-deficient population (defined by high HRD score and/or germline or somatic *BRCA1*/*2* mutation) [85]. Although negative for its primary endpoint, this study reported better tolerance and a very promising pCR rate with paclitaxel + olaparib (55%). Interestingly, subgroup analyses failed to show any difference in pCR rates between olaparib and platinum in *BRCA*-mutated patients or in *BRCA* WT HRD-high subgroups of patients [85].

PARP inhibitors have also been tested as monotherapy before chemotherapy in TNBC in window of opportunity (WOO) trials. For example, the RIO study tested rucaparib exposure for 2 weeks before surgery or NAC, with a drop in Ki67 on the end-of-treatment biopsy as primary activity endpoint. HRD tumors were identified thanks to the HRDetect tool, and there was no association between Ki67 decrease and *BRCA* mutation status, nor was there any association with HRD-high status [86]. In the phase II PETREMAC study, patients with TNBC received olaparib for 10 weeks before NAC, and 56% of patients obtained an objective response. Interestingly, contrary to non-responders, most of the responders harbored various genomic alterations potentially leading to HRD-high status, other than g*BRCA1*/*2* mutations (somatic or germline mutations of other genes involved in HR and *BRCA* promoter methylation). Moreover, functional HR deficiency assessed by low RAD 51 foci was also related to response to olaparib, contrary to *BRCA*ness signature obtained by multiplex ligation-dependent probe amplification (MLPA) [87]. This study is in favor of the activity of PARPi in TNBC beyond g*BRCA* mutations alone.

In summary, neoadjuvant and adjuvant trials and studies that have assessed response to DNA-damage therapy, according to the presence or absence of genomic instability in the setting of early breast cancer are listed in Table 1.

All in all, while the identification of g*BRCA1*/*2* mutation is no longer debated to guide the prescription of adjuvant PARPi treatment (olaparib) for patients with clinico-pathological factors of high recurrent risk, it currently remains difficult to integrate other biomarkers of HRD into treatment decisions in routine clinical practice, especially when deciding whether or not to prescribe platinum-based chemotherapy. Moreover, apart from the fact that a deficiency in the HR pathway can help to predict the response to standard NAC, many uncertainties remain in TNBC, and even more so in other subtypes—such as ER+—which have not been widely studied [29,88,89]. Nevertheless, more and more data are emerging regarding *BRCA*-mutated tumors and *BRCA*ness, and perhaps in the future, it will be possible to use these HRD biomarkers more easily to predict tumor sensitivity to neoadjuvant or adjuvant DNA-damaging agents (PARPi and platinum salts). Moreover, several ongoing phase III trials such as PEARLY (NCT02441933) and PARTNER (NCT03150576) could be practice-changing, and may thus broaden the utilization of genomic tests. This perspective raises an exciting challenge for medical oncologists and oncogeneticists.

**Table 1 cancers-15-01299-t001:** Synthesis of trials and studies evaluating genomic instability and response to DNA-damaging treatments (platinum salts and PARPi) in early conditions.

Clinical trials					
Trial Name	Phase	Stage and Subtype	Treatment	HRD Status or Condition	Main Results
**Neoadjuvant Platinum Regimen**					
GeparSixto [66]	II	Stage II-III TNBC, HER2+/ER-, HER2+/ER+ (*n = 595*)	Paclitaxel + nonpeg. lipos. doxorubicin vs. carboplatin + PM - TNBC : + bevacizumab - ER-/HER2+ : + trastuzumab + lapatinib	Among TNBC all comers : - 70.5% HRD (HRD score ≥42 or t*BRCA1*/*2* mutation) - 29% t*BRCA1*/*2* mutation - 20% g*BRCA1*/*2* mutation	- Higher pCR rate with additional carboplatin - Longer DFS with Cb (*p = 0.02*) irrespective of *BRCA* status, trend towards better OS (n.s.) - Regarding pCR : no carboplatin benefit among g*BRCA*, carboplatin benefit among *BRCA* WT - HRD predicts pathological response but does not predict carboplatin benefit - Supports clinical utility of RAD51 assay (FFPE functional HRD assay) : concordant with HRD genomic score, identifies non-t*BRCA* with functional HRD phenotype, predicts pCR and carboplatin benefit - Not a standard NAC regimen (nonpegylated liposomal anthracycline)
Byrski et al. [64]	II	Stage I-III, g*BRCA1* mutation HER2- (77% TNBC, 16% ER+) (*n = 107*)	Cisplatin	- 100% g*BRCA1* mutation	- 61% pCR rate : 61% in TNBC, 56% in ER+ - Evidence of a single platinum agent activity in g*BRCA1* mutation - Not a standard NAC regimen (anthracycline-free), no randomized control
TBCRC 008 [90]	II	Stage II-III, HER2- (39% TNBC, 61% ER+) (*n = 62*)	Carboplatin + nab-paclitaxel vs. carboplatin + nab-paclitaxel + vorinostat	Among non t*BRCA1*/*2* mutation : - 46% HRD (HRD score ≥ 42)	- 27% pCR (similar with or without vorinostat) - Small effective, not a standard NAC regimen (anthracycline-free regimen)
Kaklamani et al. [91]	II	Stage I-III, TNBC (*n = 30*)	Carboplatin + eribulin	Among TNBC all comers : - 46% HRD (HRD score ≥ 42 or g*BRCA1*/*2* mutation) - 10% g*BRCA1*/*2* mutation	- 43% pCR : 67% in g*BRCA1*/*2* mutation, 75% in HRD, HRD score and HR deficiency associated with pCR (also in *BRCA* WT population) - Small effective, not a standard NAC regimen (anthracycline-free), no randomized control
INFORM [65]	II	Stage I (≥1.5 cm)-III, g*BRCA1*/*2* mutation HER2- (70% TNBC, 30% ER+) (*n = 118*)	Cisplatin vs. AC	- 68% g*BRCA1* mutation - 30% g*BRCA2* mutation - 2% g*BCRA1* + *2* mutation	- No higher pCR rate with cisplatin in both TNBC and ER+ - Does not suppot the use of a single platinum agent regimen in g*BRCA1*/*2* mutation
TBCRC 030 [82]	II	Stage I (≥1.5 cm)-III, TNBC *BRCA* WT (*n = 147*)	Cisplatin or paclitaxel	Among *BRCA* WT : - 71% HRD positive (cut-off ≥ 33)	- HRD-score does not predict pathological response with single CT (RCB-0/1), does not support the use of HRD-score in the setting of a single platinum or taxane NAC regimen - Poor responder rate, does not support such a single NAC regimen in TNBC *BRCA* WT
**Neoadjuvant platinum-PARPi regimen**					
PreCOG 0105 [92]	II	Stage I-III, TNBC (97%) or g*BRCA1*/*2* mutation (3% ER+) (*n = 93*)	Carboplatin + gemcitabine + iniparib → surgery → AC	Among 97% TNBC and 3% HR+/HER2- : - 24% g*BRCA1*/*2* mutation	- 36% pCR : 33% in *BRCA* WT, 47% in g*BRCA1*/*2* mutation, HRD-LOH scores associated with pCR - Small effective, not a standard NAC regimen (anthracycline-free), no randomized control
I-SPY-2 [74]	II	Stage II-III, TNBC, ER+/HER2− (*n = 116*)	Paclitaxel →AC vs. paclitaxel + veliparib + carboplatin → AC	Among TNBC all comers : - 17% g*BRCA*	- 51% estimated probability of pCR rate with carboplatin-veliparib, need for results of the phase III NCT02032277
BrighTNess [67]	III	Stage II-III, TNBC all comers (*n = 634*)	Paclitaxel → AC vs. paclitaxel + carboplatin → AC vs. paclitaxel + carboplatin + veliparib → AC	Among TNBC all comers : - 67% HRD (HRD score ≥ 42 or t*BRCA1*/*2* mutation) - 15% g*BRCA1*/*2* mutation	- Higher pCR rate with additional Cb, no benefit from veliparib addition - Longer EFS with Cb (*p = 0.02*) irrespective of *BRCA* status, no difference in OS - Regarding pCR : no carboplatin benefit among g*BRCA*, carboplatin benefit among *BRCA* WT - HRD predicts pathological response but does not predict carboplatin benefit
GeparOLA [85]	II	Stage I-III, HRD-population HER2- (73% TNBC, 23% ER+) (*n = 106*)	Paclitaxel-olaparib EC → EC *or* paclitaxel-carboplatin → EC	Among HRD population : - 54% t*BRCA1*/*2* mutation - 56% g*BRCA1*/*2* mutation	- 55% pCR with PO but a potential lower rate not statistically excluded : not strong enough to change practice - Evidence of paclitaxel-PARPi combination efficacy in HRD-population with better safety
**Neoadjuvant PARPi regimen**					
NeoTALA [75]	II	Stage I (≥1 cm)-III, g*BRCA1*/*2* mutation HER2- (75% TNBC, 25% ER+) (*n = 20*)	Talazoparib	- 80% g*BRCA1* mutation - 20% g*BRCA2* mutation	- 49% pCR - Evidence of a single PARPi agent activity in g*BRCA1*/*2* mutation - Small effective, no randomized control
RIO [86]	II	TNBC (*n = 43*)	Rucaparib before surgery or NAC	- 69% HRD (HRDetect assay) - 19% g*BRCA1*/*2* mutation	- Decrease Ki67 in 12% of *BRCA* WT tumors - No association between Ki67 drop and *BRCA* mutation status, nor with HRD - Association between Ki67 drop and early ctDNA decrease - Small effective, no randomized control
PETREMAC [87]	II	Stage II-III, TNBC (*n = 32*)	Olaparib before NAC	Among TNBC all comers : - 34% HRD (g*BRCA1*/*2* and *PALB2* or somatic HR mutations) - 14% g*BRCA1*/*2* and *PALB2* mutation	- 56% OR - Higher clinical response in HRD patients and/or *BRCA1* hypermethylation and also in functional HRD harboring low RAD51 foci - Evidence of a single PARPi agent activity beyond *gBRCA* mutations - Small effective, no randomized control
Spring et al. [76]	I	Stage I (≥1 cm)-III, g*BRCA1*/*2* mutation HER2- (71% TNBC, 29% ER+) (*n = 21*)	Niraparib	- 67% g*BRCA1* mutation - 28% g*BRCA2* mutation - 5% g*BCRA1* + *2* mutation	- 40% pCR - Evidence of a single PARPi agent activity in g*BRCA1*/*2* mutation - Small effective
**Adjuvant PARPi regimen**					
OlympiA [77]	III	g*BRCA1*/*2* mutation with high risk HER2- (82% TNBC, 18% ER+) (*n = 1836*)	Olaparib	- 72% g*BRCA1* mutation - 27% g*BRCA2* mutation - <1% g*BCRA1* + *2* mutation	- Longer iDFS and OS with olaparib - Strong evidence supporting HR gene analysis of *BRCA* in this setting

Adjuvant/neoadjuvant trials including platinum salts alone are shown in blue, those with PARP inhibitors alone in green, and those with platinum salts and PARP inhibitors in orange. *n* = number of patients included. HR : homologous recombination; HRD: homologous recombination deficiency; TNBC: triple negative breast cancer; g*BRCA1*/*2* mutation: germline *BRCA1*/*2* mutation; t*BRCA1*/*2* mutation: tumor *BRCA1*/*2* mutation; *BRCA* WT: *BRCA* wild type; ER−: estrogen receptor negative cancer; ER+: estrogen receptor positive cancer; HER2−: HER2-negative cancer; HER2+: HER2-positive cancer; pCR: pathological complete response; RCB: residual cancer burden; DFS: disease free survival; iDFS: invasive disease free survival; Cb: carboplatin; FFPE: formalin fixed paraffin embedded; CT: chemotherapy; PARPi: poly ADP ribose polymerase inhibitor; LOH: loss of heterozygosity; EFS: event-free survival; OS: overall survival; NAC: neoadjuvant chemotherapy; AC: doxorubicin-cyclophosphamide; EC: epirubicin-cyclophosphamide; PM: paclitaxel + nonpegylated liposomal doxorubicin; PO: paclitaxel + olaparib; ctDNA, circulating tumor deoxyribonucleic acid; n.s.: non-significant.

## 3. Metastatic Breast Cancer (mBC)

In the metastatic setting, fewer data are currently available, especially for patients with metastatic non-TNBC subtypes. Nevertheless, because quality of life is a major concern in the metastatic setting, there is a compelling need for biomarkers that predict sensitivity to drugs such as platinum salts or PARPi. Given that these drugs have a number of side effects, such assays would be helpful to ensure that prescription is pertinent. As in the localized setting, we distinguish *BRCA* mutations from mutations in other genes involved in HR (*BRCA*ness condition).

### 3.1. Platinum and PARPi in BRCA1/2-Mutated mBC

In 2012, Byrski et al. evaluated the efficacy of cisplatin chemotherapy in *BRCA1* mutation carriers with mBC [93]. In the phase II study, only 20 patients were included: 9 of them had previously been treated for metastatic disease with at least two lines of therapy; 30% were ER+/HER2− and 70% had TNBC. The overall response rate was 80%. Overall survival was 80% at one year, 60% at two years, and 25% at three years, with a median time to progression of 12 months. This study was one of the first to demonstrate the value of platinum in advanced metastatic disease, in the presence of genomic instability represented by the *BRCA1* mutation. Several years later, the phase III TNT trial randomly assigned patients with metastatic TNBC to either docetaxel or carboplatin in the first line of treatment [94]. Results showed that carboplatin was associated with a significantly higher overall response rate (68% vs. 33%, *p* = 0.03) and improved progression-free survival (6.8 vs. 4.4 months, *p* = 0.002) for the 43 g*BRCA* mutation carriers enrolled, in contrast to those without *BRCA* mutation.

More recently, PARP inhibitors have also emerged in the treatment of mBC, primarily in g*BRCA*-mutated patients. The OlympiAD trial was designed to compare the efficacy and safety of olaparib versus the standard single-agent chemotherapy of the physician’s choice among patients with HER2-negative mBC and a g*BRCA1*/*2* mutation [80]. Olaparib monotherapy provided a significant benefit over standard therapy; median PFS was 2.8 months longer (7.0 months vs. 4.2 months; HR = 0.58; 95% CI, 0.43 to 0.80) and the risk of disease progression or death was 42% lower with olaparib monotherapy than with standard therapy. The response rate in the olaparib group was approximately twice that of the standard-therapy group (59.9% vs. 28.8%). An exploratory analysis conducted in nearly half of the overall study population showed strong concordance (99%) between g*BRCA* and t*BRCA* mutation. PARPi efficacy was similar, irrespective of HRD score, suggesting that there may be no need for additional tumor testing in case of g*BRCA1*/*2* mutation in the decision-making process [95]. In the phase II ABRAZO trial, talazoparib also showed promising activity in two cohorts of patients with mBC and g*BRCA1*/*2* mutation [96]. The response rate was 21% among patients who had previously had response to platinum chemotherapy. Then, in the phase III EMBRACA trial, patients with mBC and g*BRCA1*/*2* mutation were assigned to receive talazoparib or a standard single-agent chemotherapy of the physician’s choice [97]. The risk of disease progression or death was 46% lower in the talazoparib group than in the standard-therapy group (HR = 0.54; 95% CI, 0.41 to 0.71), with a doubling of the response rate (62.6% in the talazoparib group vs. 27.2% in the standard-therapy group). Moreover, clinical benefit obtained with these single PARPi regimens was observed irrespective of g*BRCA* mutation type (g*BRCA1* or g*BRCA2*) or BC subtype (TNBC or ER+). The results of the two phase III trials (OlympiAD and EMBRACA) led to the approval of olaparib and talazoparib for the treatment of mBC with g*BRCA1*/*2* mutation, and international guidelines now recommend the systematic testing of patients with ER+/HER2− or triple negative mBC, in order to enable early treatment of these patients with PARPi during their metastatic history [98].

Combining DNA-damage therapies was later assessed in BROCADE3, a randomized, phase III trial that tested the association of veliparib with carboplatin and paclitaxel in *BRCA*-mutated advanced breast cancer [99]. Patients were randomly assigned to carboplatin and paclitaxel plus veliparib (veliparib group) or carboplatin and paclitaxel plus placebo (control group). Median PFS was 14.5 months in the veliparib group versus 12.6 months in the control group (HR = 0.71 [95% CI 0.57–0.88], *p* = 0.0016). The addition of veliparib to a highly active platinum doublet—with continuation as monotherapy if the doublet were discontinued—resulted in a significant and durable improvement in PFS in patients with g*BRCA*-mutated advanced HER2-negative breast cancer. These data may indicate the utility of combining platinum and PARP inhibitors in this *BRCA*-mutated metastatic population, particularly as continuation therapy.

There is therefore a rationale for the use of platinum salts and PARP inhibitors in g*BRCA* mutated patients early in the course of metastatic disease. However, in the vast majority of these studies, HRD score was not evaluated. Thus, patients with potential genomic instability without g*BRCA* mutation were not included. 

### 3.2. Platinum and PARPi beyond BRCA-Mutated mBC

In mTNBC treated in first or second line with platinum monotherapy, the TBCRC 009 phase II trial evaluated the objective response rate (ORR) in 86 patients, according to their *BRCA* and HRD status. In this study, Isakoff et al. reported a response rate of 25.6% in the overall population, and a higher rate (54.5%) in patients with g*BRCA1*/*2* mutations (*n* = 11) [100]. In patients without *BRCA1*/*2* mutation, exploratory analyses conducted on 32 patients showed higher HRD features (high LST and LOH scores) in responding patients. These pioneering data suggest that some HRD-derived biomarkers may help to preferentially choose a platinum salt early in the disease course; As previously described, beyond *BRCA1*/*2* mutations, many other genomic and epigenetic alterations may explain the inactivation of different HR components, leading to HRD in *BRCA*-proficient tumors (so-called *BRCA*ness phenotype). However, in the TNT trial, no benefit of carboplatin over docetaxel was observed in mTNBC patients with *BRCA 1* methylation, *BRCA*1 mRNA-low tumors, or in patients whose tumor harbored other HRD features, such as a high HRD score (by the Myriad assay) [94]. Indeed, a high HRD score was associated with an ORR of 44.7% with carboplatin versus 39.6% with docetaxel (*p* = 0.67). Similarly, no evidence of an increase in median PFS was observed in high-HRD versus non-HRD tumors. 

Using the aforementioned HRDetect assay (based on WGS), but in metastatic conditions, Zhao et al. found that an elevated HRDetect score was significantly associated with response to platinum-based chemotherapy in a small series of mBC patients [101]. Thus, although the TNT trial did not find any association between HRD score and response to platinum, Zhao’s results re-open the debate in metastatic HRD-high *BRCA* proficient patients, also regarding the best technique for assessing HRD (commercial tools or WGS). These exploratory results will need to be confirmed in prospective trials.

Galland et al. evaluated response to platinum and survival in 86 patients with mBC of any subtype (50% ER+) [102] and multi-treated (>60% had received three or more prior lines of therapy). Using WES for the determination of the HRD score or the COSMIC signature 3 expression, patients were classified into three groups: *BRCA*-mutated, *BRCA* WT HRD-high (or S3 high), and *BRCA* WT HRD-low (or S3 low). As in Zhao’s study mentioned above, Galland et al. were able to identify a subset of *BRCA* WT mBC harboring high HRD scores (≥42) and a high S3 mutational signature, at levels comparable to those of *BRCA1*/*2* mutated tumors [101]. However, in this study, the mBC patients with high HRD score or high S3 level did not seem to benefit more from platinum-based chemotherapy than the others, in terms of response and/or PFS, regardless of BC molecular subtype and HRD or S3 cut-off. This study was one of the first to look at subtypes other than TNBC for the determination of HRD-associated genomic features. Indeed, these results were also in accordance with recent publications conducted in a large cohort of BC patients with WGS approaches, showing that HRDetect high scores were also observed in ER+ tumors. Similarly, in a recent, large-scale genomic characterization of mBC, Bertucci et al. reported both increased somatic genomic alterations in genes involved in HR pathway, and more HRD features (e.g., increased S3 mutational signature) in mBC, as compared to eBC, particularly in the ER+/HER2− subtype [103]. This highlights the need to look at subtypes other than TNBC in the study of biomarkers of HRD and sensitivity to treatments such as platinum salts.

Trials and studies that have already looked at mBC and response to platinum salts and PARP inhibitors according to the presence of genomic instability are listed in Table 2.

**Table 2 cancers-15-01299-t002:** Synthesis of trials and studies evaluating genomic instability and response to treatments (platinum salts and PARPi) in mBC (early or advanced metastatic condition).

Clinical Trial						
Trial Name	Stage	Line	Subtype	Treatment	HRD Status or Condition	Main Results
**Metastatic platinum regimen**						
TBCRC009 [100]	IV	Early metastatic condition (*n = 86*)	TNBC	Carboplatin or Cisplatin	*BRCA1*/*2* mutation and HRD score (HRD-LST and HRD-LOH)	- Patients with *BRCA1*/*2* mutation : ORR >50% (vs. 25% in total population) - In patients without *BRCA1*/*2* mutation, higher HRD scores in responding patients
TNT [94]	III-IV	Early metastatic condition (*n = 376*)	TNBC	Carboplatin vs. Docetaxel	*BRCA1*/*2* mutation (germline or somatic) or *BRCA1* hypermethylation or HRD score > 42	- Significantly higher ORR and PFS for the g*BRCA* mutation carriers - No benefit in patients *with BRCA1* hypermethylation or in patients whose tumor harbored a high HRD score
Byrski et al. [93]	IV	Advanced metastatic condition (*n = 20*)	TNBC, ER+/HER2−	Cisplatin	*BRCA1* mutation carriers	Interesting platinum salts efficacy in the presence of a *BRCA1* mutation
Zhao et al. [101]	IV	Early metastatic condition (*n = 33*)	TNBC, ER+/HER2− and HER2+	Carboplatin or Cisplatin	HRDetect status (WGS)	Radiographic evidence of clinical improvement, and better survival and treatment duration in patients with high HRDetect and treated with platinum salts
Galland et al. [102]	IV	Early and advanced metastatic condition (*n = 86*)	TNBC, ER+/HER2− and HER2+	Carboplatin or Cisplatin	HRD score and COSMIC signature 3 (WES)	- Subset of *BRCA*-proficient tumors with high HRD score or high S3 levels, comparable to *BRCA*-mutated tumors - However, no better ORR/DCR and PFS in these patients treated with platinum salts than the others
**Metastatic platinum-PARPi regimen**						
BROCADE3 [99]	III-IV	Early metastatic condition (*n = 513*)	TNBC	Carboplatin + Paclitaxel vs. Carboplatin + Paclitaxel + veliparib vs. Veliparib	g*BRCA1*/*2* mutation	The addition of veliparib to a highly active platinum doublet resulted in a significant improvement in PFS in patients with g*BRCA* mutation
**Metastatic PARPi regimen**						
OlympiAD [80]	IV	Early metastatic condition (*n = 302*)	TNBC, ER+/HER2−	Olaparib	g*BRCA1*/*2* mutation	- Significant benefit over standard therapy in g*BRCA* carriers - Benefit irrespective of g*BRCA* mutation type (g*BRCA1 or 2*), of BC subtype (TNBC and ER+) and of HRD score
EMBRACA [97]	IV	Early metastatic condition (*n = 431*)	TNBC, ER+/HER2−	Talazoparib	g*BRCA1*/*2* mutation	- Significant benefit over standard therapy in g*BRCA* carriers - Benefit irrespective of g*BRCA* mutation type (g*BRCA1* or *2*) and of BC subtype (TNBC and ER+)
TOPACIO [104]	IV	Early and advanced metastatic condition (*n = 55*)	TNBC	Niraparib + pembrolizumab	*gBRCA1/2* mutation or *BRCA1*/*2* WT	- ORR = 25% among the 60 *BRCA*1/2 WT and ORR = 45% among the 11 *BRCA1*/*2*-mutated tumors - Promising antitumor activity, irresepectively of *BRCA* mutation in mBC
MEDIOLA [105]	IV	Early and advanced metastatic condition (*n = 34*)	TNBC, ER+/HER2−	Olaparib + durvalumab	g*BRCA1*/*2* mutation	- Promising antitumour activity in *gBRCA1/2*-mutated mBC
TBCRC 048 [28]	IV	Early and advanced metastatic condition (*n = 54*)	TNBC, ER+/HER2−	Olaparib	Germline mutations in non-*BRCA1*/*2* HR-related genes or t*BRCA1*/*2* mutations	- g*PALB2* : ORR = 82%, t*BRCA1*/*2* : ORR = 50%, no confirmed response among other mutation profils - Promising antitumour activity beyond g*BRCA1*/*2* mutation
RUBY [106]	IV	Early and advanced metastatic condition (*n = 42*)	TNBC, ER+/HER2−	Rucaparib	High LOH score or non-g*BRCA1*/*2* mutation	- CBR = 13.5% - Potential benefit among a small subset of patients with high LOH scores without g*BRCA1*/*2* mutation
Gruber et al. [107]	IV	Early and advanced metastatic condition (*n = 13*)	TNBC, ER+/HER2−	Talazoparib	*BRCA* WT with mutation in HR-associated gene	- ORR 31% - HRD score correlated with response : driven by g*PALB2* mutation - Promising antitumour activity beyond g*BRCA1*/*2* mutation
NCT03685331 (HOPE trial)	III-IV	Early metastatic condition	ER+/HER2−	Palbociclib + Olaparib and Fulvestrant	g*BRCA1*/*2* mutation	In progress (recruiting)
NCT04053322 (DOLAF trial)	III-IV	Early and advanced metastatic condition	ER+/HER2−	Durvalumab + Olaparib and Fulvestrant	g/t*BRCA1*/*2* mutation or HR-defect	In progress (recruiting)
NCT03025035	III-IV	Advanced metastatic condition	TNBC, ER+/HER2− HER2+	Pembrolizumab + Olaparib	g*BRCA1*/*2* mutation or HR-defect	In progress (recruiting)

Trials including platinum salts alone in metastatic conditions are shown in blue, those with PARP inhibitors alone in green, and those with platinum salts and PARP inhibitors in orange. *n*= number of patients included. HR: homologous recombination, HRD: homologous recombination deficiency, LOH: loss of heterozygosity, TNBC: triple negative breast cancer, g*BRCA 1*/*2* mutation: germline *BRCA 1*/*2* mutation, t*BRCA1*/*2* mutation: tumor *BRCA1*/*2* mutation, *BRCA* WT: *BRCA* wild type, ER+: Eostrogen receptor positive cancer, PFS: progression free survival; CBR: clinical benefit rate, ORR: objective response rate, WGS: whole genome sequencing, WES: whole exome sequencing, DCR: disease control rate.

Despite the extensive development of PARP inhibitors, they are not currently authorized for use in breast cancer outside of g*BRCA* mutations, despite the promising results reported in these patients. For example, in the TBCRC 048 trial, PARPi were shown to be effective in patients with g*PALB2* or s*BRCA1*/*2* mutations, significantly expanding the potential target population of patients with BC likely to benefit from PARPi, other than g*BRCA1*/*2* mutation carriers [28]. The RUBY study also suggested that a small subset of patients with high LOH scores without g*BRCA1*/*2* mutation may benefit from PARP inhibitors [106]. Recently, talazoparib demonstrated promising activity in 13 patients pre-treated *BRCA* WT mBC harboring a HR mutation (11 ER+ and 2 TNBC) with overall response and clinical benefit rates of 31% and 54%, respectively [107]. In this phase II study, higher HRD score was correlated with better response, mainly driven by g*PALB 2* carriers. These encouraging results open the way for PARPi treatment beyond g*BRCA1*/*2* mutation. In part for these reasons, Keung et al. studied the inhibitory activity of PARPi on various breast cancer cells, and demonstrated differential inhibitory activities independently of the *BRCA* status [108]. These results suggest that the status of *BRCA* is not the only biomarker of response to PARPi. However, many clinical trials recruit patients based on their *BRCA* mutation status and do not incorporate HRD testing or *BRCA*ness phenotype. Furthermore, in order to expand the potential prescription of PARPi to *BRCA* WT patients with genomic instability and to better identify patients likely to respond to such treatments, efforts are under way to develop new technologies. McGrail et al. generated a novel predictive algorithm able to predict PARPi response in different cell lines and patient-derived tumor cells [109]. This PARPi sensitivity signature could serve as an important tool to identify patients without *BRCA* mutation, but with HR defects and *BRCA*ness phenotype. Through the integration of novel HRD biomarkers and scoring systems, the identification of patient populations who may have therapeutic sensitivity to PARPi may be an advantage in mBC. However, this will require confirmation in future clinical trials and is not currently recommended. 

Due to the increasing importance of immunotherapy (immune checkpoint inhibitors, ICI) in oncology, there is increasing focus on the rationale for combining immunotherapy and PARP inhibitors in *BRCA*-mutated tumors, but also in tumors with *BRCA*ness as proposed in the DOLAF study (NCT04053322). This study, which is currently recruiting, aims to evaluate the efficacy of a combination of olaparib, durvalumab, and fulvestrant for the treatment of patients with locally advanced or metastatic breast cancer with *BRCA* mutation or alterations of genes involved in HRR. Concerning the rationale for adding immunotherapy in case of HR deficiency, Mao et al. demonstrated that tumors with an S3 mutational signature had high expression of certain checkpoint inhibitors of the immune response, such as CTLA-4 or PD-L1 [110]. Teo et al. also suggested that mutations in HR pathways may positively influence response to ICI [111]. Thus, the combination of immunotherapy with PARPi appears attractive and has yielded encouraging initial clinical results in *BRCA*-mutated tumors. The MEDIOLA trial assessed the efficacy of olaparib in combination with durvalumab in patients with g*BRCA*-mutated mBC [105]. Patients with *BRCA* WT tumors were also included, as in the TOPACIO trial, where the combination of niraparib and pembrolizumab provided promising antitumor activity, irrespective of *BRCA* mutation, with ORRs of 25% and 45% respectively among the 60 *BRCA* WT and 11 *BRCA*-mutated tumors [104]. The ORR observed in patients with *BRCA*-mutated tumors was similar to that reported with olaparib monotherapy in the OlympiAD trial. However, the median PFS of 8.3 months in these patients was nearly 3 months longer than that observed for olaparib (5.6 months) or talazoparib (5.8 months) in patients with g*BRCA-*mutated TNBC. The few good responses observed among *BRCA* WT patients raise questions about the presence of other mutations in the homologous recombination pathway. To further elucidate this issue, an ongoing trial (NCT03025035) evaluating the combination of pembrolizumab plus olaparib will focus on this population by including *BRCA* WT patients with HRD.

In total, the identification of *BRCA*ness status by mutations other than *BRCA*, or the determination of HRD score, could provide benefit to a significant number of patients by enabling the prescription of PARPi and the enrolment in therapeutic trials combining such treatments with immunotherapy. This represents a major challenge for the future.

### 3.3. Limitations in the Use of HRD Biomarkers in the Metastatic Condition

The current lack of consensus highlights the need for further evaluation of the role of HRD biomarkers, as well as the need for methodological optimization to properly ascertain HRD high tumors. Furthermore, despite the optimization of HRD determination, it is important to take into account that HRD status is likely to change during the course of metastatic disease. This may contribute to the different results observed when studying therapeutic response and survival according to HRD biomarkers in early or advanced stages of metastatic disease.

First, the majority of patients treated in these studies received adjuvant treatment with agents that cause DNA damage, engaging the homologous recombination system. Ter Brugge et al. showed relevant resistance mechanisms to double-strand break DNA drugs (e.g., cisplatin, melphalan, or olaparib) on a cohort of 75 mice carrying *BRCA*1-deficient (mutated or promoter hypermethylation) breast tumors [112]. A number of *BRCA 1*-methylated tumors acquired therapy resistance via re-expression of *BRCA* 1 because of the loss of *BRCA1* promoter methylation. It is postulated that *BRCA* methylated tumors treated with adjuvant or neoadjuvant chemotherapy could modify their genetic functionality during treatment since they continue to express the alterations contributing to the HRD score, but drive the tumors towards a soft *BRCA*ness phenotype. An interesting example comes from ovarian cancer, where *BRCA* mutation was found to be related to platinum response, in contrast to tumors with hypermethylation of the *BRCA* promoter [113]. To explore this phenomenon, a tumor biopsy was obtained before and after platinum treatment and showed a reversal of *BRCA 1* methylation in 31% of tumors [114,115]. The genome evolves during the metastatic process and is correlated with an increase in the percentage of genomic scars previously associated with HRD [103]. However, these biological tests, based on the study of genomic scars, do not take into account the potential restoration of functional homologous recombination (which is a resistance mechanism that can appear under therapeutic pressure) [23]. Indeed, a genomic analysis conducted in a *gBRCA1-*mutated patient who had poor response to a NAC platinum-containing regimen with early metastatic relapse and death demonstrated the existence of a reverse *BRCA 1* mutation arising between the original breast tumor and the residual surgical tissue. This led to restored *BRCA 1* function that could have explained the chemoresistance [116]. Moreover, *BRCA* status analysis performed at recurrence found the same mutation on metastatic tissue. In addition, subgroup analyses performed in Olympia [77] for eBC, and in OlympiAD [80] and EMBRACA [97] for mBC, suggest that PARPi may yield less benefit in patients pre-treated with platinum. Altogether, these findings raise questions about the therapeutic sequence with DNA-damage therapies that could give rise to resistance mechanisms, especially when platinum salts are followed by PARPi. It would therefore be useful to incorporate functional biomarkers, such as evaluation of *RAD 51* foci, as a predictive biomarker of functional HR. As previously described, RAD 51 nuclear foci is a surrogate marker of HRR functionality. Cruz et al. reported that the detection of RAD 51 foci in g*BRCA* tumors correlates with PARPi resistance, regardless of the underlying mechanism restoring HRR function [46].

A further question is that of the tissue on which the assessment of homologous recombination functionality is performed; namely, whether it should be on the primary tumor or on metastasis. Indeed, there are biological differences that make it difficult to extrapolate the analysis of homologous recombination from a localized situation to a metastatic situation. These findings suggest that the HRD assay is promising in concept, but whether it can be used to identify somatic or *gBRCA* WT patients who may benefit from PARPi or platinum-based therapy remains to be determined.

Lastly, because of the complexity of the homologous recombination phenomenon and progress in knowledge about it, increasingly complex methods are being used to develop tools and scores likely to predict the effectiveness of treatments targeting DNA. For this reason, the application of WGS, including the HRDetect score for example, in clinical practice is a controversial topic, given the financial costs. Indeed, it is necessary to consider the large number of patients with breast cancer around the world. Moreover, HRD status can vary during the history of a patient’s disease, so the question arises of the best timing (localized/early stage metastatic/late stage metastatic), in order to limit potential multiplication of these analyses, and consequently, the costs incurred. In the same manner, acquisition and analysis of WGS-based data calls for large and complex sequence analysis, requiring considerable bioinformatics expertise and associated with technical issues. Altogether, obtaining a HRDetect score represents a limitation to daily clinical practice at the present time. However, the steady drop in the cost of sequencing could make more widespread use of WGS possible in years to come.

## 4. Conclusions

With the help of next-generation sequencing, the development of biomedical technologies and the use of bioinformatics, it is now possible to identify specific molecular alterations, such as HR deficiencies, which make it possible to consider effective targeted drugs. It appears that the clinical utility of genomic biomarkers assessing HRD in breast cancer is more moderate than in ovarian cancer, with sometimes discordant results, as in metastatic disease. Currently, only the identification of a germline mutation in the *BRCA 1* or *2* gene guides the use of platinum salts (only in the metastatic setting) and PARP inhibitors (both in the adjuvant and metastatic settings), with several clinical approvals (olaparib, talazoparib). The value of mutations in other genes involved in the homologous recombination pathway (e.g., *RAD 51C*, *PALB 2*, *RAD 51D*), genomic scar or mutational signatures (e.g., HRD score, COSMIC signature 3, 8), or functional tests (RAD 51 foci) in guiding the use of specific therapies remains debated. Nevertheless, there is growing consensus that it is now possible to identify patients who respond to platinum salts or PARPi using these different scores. For this reason, patients with a *BRCA*ness profile need to be included in greater numbers in future therapeutic trials, with stratification on HRD status. Finally, aside from their potential clinical utility, integrating these scores into daily practice may be challenging, since their routine use will require technical competence and financial resources.

## Figures and Tables

**Figure 1 cancers-15-01299-f001:**
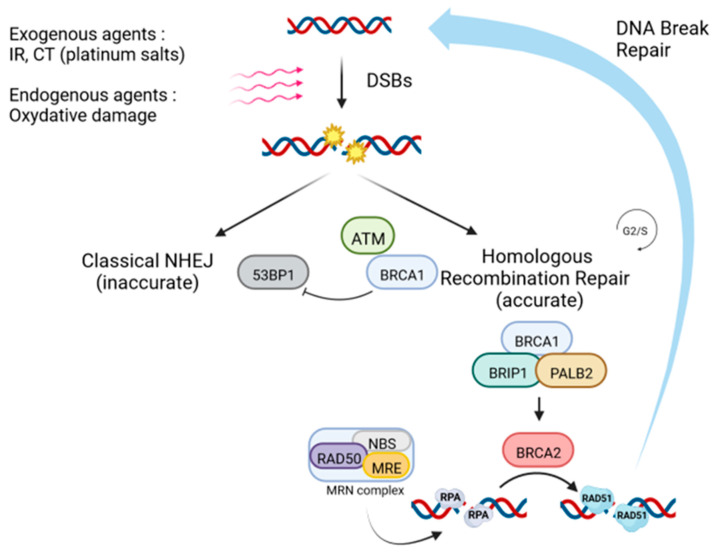
Homologous recombination repair pathway (made with Biorender). Repair begins at DSB sites by recruitment of ATM, which phosphorylates proteins such as BRCA1 in the case of HRR, or 53BP1 for NHEJ. Focusing on the HRR pathway, activation of BRCA1, via BRCA2 and PALB2, enables the transformation of DSBs into single-stranded DNA, to which the RPA proteins hybridize. This step also involves the MRN complex. RAD51 will then replace the RPA proteins bound to the single-stranded DNA, enabling the search for homology sequences, and involvement in strand invasion. The last step consists of DNA synthesis, ligation, and resolution of the Holliday junctions. IR: ionizing radiation, CT: chemotherapy, DSBs: DNA double-strand breaks, NHEJ: non-homologous end joining, DNA: deoxyribonucleic acid.

**Figure 2 cancers-15-01299-f002:**
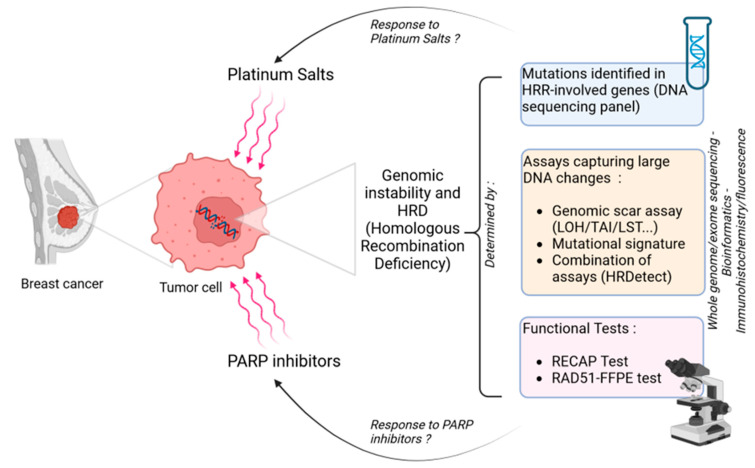
Homologous recombination deficiency (HRD) evaluation in breast cancer to predict platinum salts or PARP inhibitors response (made with Biorender). HRR: Homologous recombination repair, HRD: homologous recombination deficiency, DNA: deoxyribonucleic acid, LOH: loss of heterozygosity, TAI: telomeric allelic imbalance, LST: large-scale state transitions, PARP: poly(ADP-ribose) polymerase.

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
