# Peer review of "Clinical Utility of Genomic Tests Evaluating Homologous Recombination Repair Deficiency (HRD) for Treatment Decisions in Early and Metastatic Breast Cancer"

_cancers, 2023, doi:10.3390/cancers15041299_

Round 1
Reviewer 1 Report
In the review paper by Galland et al., the authors described the processes of HRD, assays measuring HRD and clinical trials evaluating platinum agents/PARPi in patients with HRD tumors. This was a timely review and a very interesting read. I look forward to the publication and will include it as a citable entry for my own future manuscripts.
I have only a few minor suggestions:
Gene names should be italisized. The authors do this a few times, with BRCA but are not consistent and did not italisize most other gene names.
There are multiple instances in which the authors include lists which parentheses and end with … (X, Y, ….). Please either replace the … with another item or just remove.
There are some English issues throughout, starting with the second sentence of the simple summary. It would be helpful to have the review read again by someone with strong English writing skills.
Author Response
Reviewer #1
In the review paper by Galland et al., the authors described the processes of HRD, assays measuring HRD and clinical trials evaluating platinum agents/PARPi in patients with HRD tumors. This was a timely review and a very interesting read. I look forward to the publication and will include it as a citable entry for my own future manuscripts.
Thank you for your positive appreciation of our work.
I have only a few minor suggestions:
- Gene names should be italisized. The authors do this a few times, with BRCA but are not consistent and did not italisize most other gene names.
Thank you for this comment. We have corrected the names of the genes that were not in italics. The proteins associated with these genes have not been italicized. Corrections are highlighted in yellow in the manuscript : for example « BRCA 1 »
- There are multiple instances in which the authors include lists which parentheses and end with … (X, Y, ….). Please either replace the … with another item or just remove.
We have corrected as suggested.
- There are some English issues throughout, starting with the second sentence of the simple summary. It would be helpful to have the review read again by someone with strong English writing skills.
We apologize for the errors encountered. The paper has been revised again by a native English-speaking medical writer with expertise in the domain. We are confident that the quality of the English is now acceptable.

Reviewer 2 Report
The review article by Galland et. al, summarizes the current knowledge regarding the clinical utility of genomic tests available evaluating the HRR (homologous recombination repair) deficiency for treatment decisions in early and metastatic breast cancer. Homologous recombination is one of the major pathways for the repair of DNA double strand breaks. An underlying cause of breast cancer has largely been attributed to the defects in the DNA damage response pathways. The authors have included recent papers by experts in the field and the article does seem to be comprehensive and balanced. The article is well written and each sub section is explained in detail with all the updated information included. overall, a very well written article.
There is a minor typo which need to be corrected:
1) Line 323-327 and 339-340, the font looks different.
Author Response
Reviewer #2
The review article by Galland et. al, summarizes the current knowledge regarding the clinical utility of genomic tests available evaluating the HRR (homologous recombination repair) deficiency for treatment decisions in early and metastatic breast cancer. Homologous recombination is one of the major pathways for the repair of DNA double strand breaks. An underlying cause of breast cancer has largely been attributed to the defects in the DNA damage response pathways. The authors have included recent papers by experts in the field and the article does seem to be comprehensive and balanced. The article is well written and each sub section is explained in detail with all the updated information included. overall, a very well written article.
Thank you for reading and commenting, and thank you for your positive appreciation.
There is a minor typo which need to be corrected:
- Line 323-327 and 339-340, the font looks different.
We have harmonized the font throughout the manuscript.

Reviewer 3 Report
This review informs about tests for DNA repair deficient breast cancers, homologous recombination repair deficiency (HRD). Those represent somatic or germline mutations. Comments are made on localized and metastatic disease.
The paper in its current state is very long and difficult to read. Sentences that express opinion are interspersed with listings of trials. While overall well referenced, the paper would benefit from more concise organization. “Indeed, there are biological differences that make it difficult to extrapolate the analysis of homologous recombination from a localized situation to a metastatic situation.” (L113-L115).
The tables with lists of clinical trials are helpful. Also, the schematic figures are visually appealing and illustrate complex pathways well. However, figure legends could describe the mechanisms more clearly. Trials could be be tabulated consistently with numbers of enrolled patients and results to allow meaningful comparison.
Abbreviations are numerous, a glossary of abbreviations might be helpful for the reader.
“…” or “… etc” is used repeatedly (L 50, L52, 63, L90, L157 and more). This conveys casual imprecision and should be replaced by content.
Extensive editing for clarity would be needed. Some sentences are difficult to understand: “Nevertheless, it is particularly in this metastatic situation, where quality of life is paramount, that the use of biomarkers of sensitivity to drugs such as platinum salts or iPARPs seems interesting. As these drugs are not without adverse effects, being able to help the clinician in prescribing them would be relevant.” (P14 1st paragraph).
Some statements resemble an opinion piece and appear without references “Lastly, because of the complexity of the homologous recombination phenomenon, 118 and our enhanced understanding of it, increasingly complex methods are being used to 119 develop tools and scores likely to predict the effectiveness of treatments targeting DNA. “ (L118-120).
What is the meaning of a sentence like this? “Finally, beyond their potential clinical utility, the routine use of these scores will undoubtedly be accompanied by technical and financial challenges in the future.” (L148-149).
Author Response
Reviewer #3
This review informs about tests for DNA repair deficient breast cancers, homologous recombination repair deficiency (HRD). Those represent somatic or germline mutations. Comments are made on localized and metastatic disease.
- The paper in its current state is very long and difficult to read. Sentences that express opinion are interspersed with listings of trials. While overall well referenced, the paper would benefit from more concise organization.
We apologize if the Reviewer found the paper difficult to read. We voluntarily organized the paper into several different parts, as best we could, with a general background section, a section about localized cancer, and a section about metastatic cancer. For these localized and metastatic sections, we distinguish between the two situations encountered, namely BRCA mutated or BRCA WT. Finally, we created tables to structure these two parts (localized and metastatic), in order to facilitate the reading and summarize the majority of the trials.
Nevertheless, as suggested by the Reviewer, we have reworded the various points specified, modified the figure legends, added numbers in the tables and created a list of abbreviations in order to satisfy your requests as much as possible.
Moreover, the paper has been revised again by a native English-speaking medical writer with expertise in the domain. We are confident that the quality of the English is now acceptable.
- The tables with lists of clinical trials are helpful. Also, the schematic figures are visually appealing and illustrate complex pathways well. However, figure legends could describe the mechanisms more clearly.
As suggested, we have clarified the legend of Figure 1.
“Repair begins at DSB sites by recruitment of ATM, which phosphorylates proteins such as BRCA 1 in the case of HRR, or 53BP1 for NHEJ. Focusing on the HRR pathway, activation of BRCA 1, via BRCA2 and PALB2, enables the transformation of DSBs into single-stranded DNA, to which the RPA proteins hybridize. This step also involves the MNR complex. RAD 51 will then replace the RPA proteins bound to the single-stranded DNA, enabling the search for homology sequences, and involvement in strand invasion. The last step consists of DNA synthesis, ligation and resolution of the Holliday junctions.”
- Trials could be be tabulated consistently with numbers of enrolled patients and results to allow meaningful comparison.
As suggested, we have summarized the trials in the tables to facilitate the reading of the review. The main results were already specified in the tables.
As requested, we have added the number of enrolled patients in Tables 1 and 2 : (n= x).
We have also specified the notation in the legends, as follows: "n=x" is the number of included patients.
- Abbreviations are numerous, a glossary of abbreviations might be helpful for the reader.
As requested, we have created a list of the abbreviations used in the review. We leave it to the editor’s discretion to incorporate the list at the appropriate place in the review.
NGS : Next Generation Sequencing
HRD : Homologous Recombination Deficiency
HRR : Homologous Recombination Repair
HR : Homologous Recombination
DSBs : DNA double strand breaks
NHEJ: non-homologous end joining
AltEJ : micro-homologous end joining
BC : Breast Cancers
TNBC : Triple Negative Breast Cancers
ER : Estrogen Receptor
IR: Ionizing Radiation
CT: Chemotherapy
PARP : Poly-(ADP-ribose) Polymerases
SSBs : Single Strand Breaks
BER : Base Excision Repair
PARPi: PARP inhibitors
WT : wild-type
LST : Large-scale state transition
TAI : Telomeric Allelic Imbalance
LOH : Loss Of Heterozygosity (LOH)
TCGA : The Cancer Genome Atlas
ICGC : International Cancer Genome Consortium “
COSMIC : Catalogue of Somatic Mutations In Cancer
SigMA : Signature Multivariate Analysis
WGS : Whole Genome Sequencing
WES : Whole Exome Sequencing
RECAP test : REcombination CAPacity test
eBC : early Breast Cancer
pCR : pathologic Complete Response
NAC : NeoAdjuvant Chemotherapy
WOO : Window Of Opportunity
MLPA : Multiplex Ligation Dependent probe Amplification
RCB: Residual Cancer Burden
DFS: Disease Free Survival
iDFS: invasive disease free survival
Cb: Carboplatin
FFPE: formalin fixed paraffin embedded
EFS: event-free survival
OS: overall survival
AC : Doxorubicin-cyclophosphamide
EC : Epirubicin-cyclophosphamide
PM : Paclitaxel + non-pegylated liposomal doxorubicin;
PO : paclitaxel + Olaparib;
ctDNA : circulating tumor Deoxyribonucleic Acid
n.s.: non significant.
mBC : metastatic Breast Cancer
ICI : Immune Checkpoint Inhibitors
ORR : Objective Response Rate
CBR: Clinical Renefit Rate
DCR : Disease Control Rate
- “…” or “… etc” is used repeatedly (L 50, L52, 63, L90, L157 and more). This conveys casual imprecision and should be replaced by content.
We have corrected this throughout the manuscript: for example : “e.g. during meiosis or telomere erosions”, “such as radiotherapy or chemotherapy”, “such as RAD51C epimutations, inactivation of PALB2, BRIP1 or BARD1”.
Otherwise, when used inappropriately, we have removed the “etc / …”
- Some sentences are difficult to understand: “Nevertheless, it is particularly in this metastatic situation, where quality of life is paramount, that the use of biomarkers of sensitivity to drugs such as platinum salts or iPARPs seems interesting. As these drugs are not without adverse effects, being able to help the clinician in prescribing them would be relevant.” (P14 1stparagraph).
We propose to rephrase as follows:
“Nevertheless, because quality of life is a major concern in the metastatic setting, there is a compelling need for biomarkers that predict sensitivity to drugs such as platinum salts or PARPi. Given that these drugs have a number of side effects, such assays would be helpful to ensure that prescription is pertinent.”
- Some statements resemble an opinion piece and appear without references “Lastly, because of the complexity of the homologous recombination phenomenon, and our enhanced understanding of it, increasingly complex methods are being used to develop tools and scores likely to predict the effectiveness of treatments targeting DNA. “ (L118-120).
This sentence summarizes what we wanted to emphasize through our review, notably: the molecular complexity of breast tumors, the complexity of the homologous recombination pathway, and the usefulness of biomarkers predictive of response to current or future therapies. The advent of new technologies such as WGS may open new perspectives, particularly in BRCA WT patients.
It is possible that the confusion stems from the use of the pronoun “our” in the sentence, which may suggest that it is a personal opinion. We have rephrased this as follows:
“Lastly, because of the complexity of the homologous recombination phenomenon and progress in knowledge about it, increasingly complex methods are being used to develop tools and scores likely to predict the effectiveness of treatments targeting DNA.”
- What is the meaning of a sentence like this? “Finally, beyond their potential clinical utility, the routine use of these scores will undoubtedly be accompanied by technical and financial challenges in the future.” (L148-149).
We have rephrased as follows :
“Finally, aside from their potential clinical utility, integrating these scores into daily practice may be challenging, since their routine use will require technical competence and financial resources.”
